# Genetic Evaluation of Bovine Papillomavirus Types Associated with Teat Papillomatosis in Polish Dairy Cattle with the Report of a New Putative Type

**DOI:** 10.3390/pathogens12111278

**Published:** 2023-10-25

**Authors:** Paulina Pyrek, Michał Bednarski, Jarosław Popiel, Magdalena Siedlecka, Magdalena Karwańska

**Affiliations:** 1Department of Internal Medicine and Clinic of Diseases of Horses, Dogs and Cats, Wrocław University of Environmental and Life Sciences, Grunwaldzki Sq. 47, 50-366 Wrocław, Poland; jaroslaw.popiel@upwr.edu.pl; 2Department of Production Animal Clinical Sciences (PRODMED), Norwegian University of Life Science, Elizabeth Stephansens vei 15, 1430 Ås, Norway; 3Department of Epizootiology with Exotic Animal and Bird Clinic, Wrocław University of Environmental and Life Sciences, Grunwaldzki Sq. 45, 50-366 Wrocław, Poland; michal.bednarski@upwr.edu.pl (M.B.); magdalena.siedlecka@upwr.edu.pl (M.S.); magdalena.karwanska@upwr.edu.pl (M.K.)

**Keywords:** bovine papillomavirus, Poland, cattle, teat, papillomatosis

## Abstract

Teat papillomatosis is reported to be one of the factors causing mastitis and milk losses in dairy cattle. Little is known about bovine papillomavirus (BPV) circulation in the European cattle population, and no reports can yet be found about its prevalence in Polish herds. In this study, 177 BPV-like lesions were collected from teats of 109 slaughtered cows. BPV was identified in 39 of the examined animals, using PCR amplification and Sanger dideoxy sequencing. In total, 10 BPV types were isolated, among which the most common were infections caused by types 8 and 7. Macroscopically, “rice-grain” type lesions dominated (76%) and were mainly found on one teat (57.4%). The diversity of BPV types causing teat papillomatosis in Polish cows seems to be large, with nine already known types isolated and a new putative type found. The spread of new types among the worldwide cattle population can be seen for the first time, as type 25 and so called isolates BPV42 and BPV43 were found in the European cattle population.

## 1. Introduction

Bovine papillomaviruses (BPVs) are double-stranded DNA viruses belonging to the *Papillomaviridae* family, widespread among the worldwide population of cattle [1]. In general, papillomaviruses are species-specific, with infections reported in mammals, birds, fish, and reptiles, and display mucosotropic, cutaneotropic or dual tropism for epithelial tissues [2]. In the last few years, the number of discovered BPV types has been slightly growing, with 29 types known nowadays and around 15 new putative ones [3,4]. Most of them belong to five genera: *Delta*-, *Epsilon*-, *Xi*-, *Dyoxi*- and *Dyokappapapillomavirus*, but fourteen (16–29) persist as non-classified [3]. Viral DNA can be found in healthy skin, as well as in papilloma, fibro-papilloma and squamous papilloma lesions of the skin, upper digestive tract, reproductive and urinary system [5,6,7,8,9].

The course of the disease is usually benign and self-limiting, but severe resistance to treatment lesions may also appear. Up to the present, nine BPV types have been reported to cause teat and udder papillomatosis (1, 3, 5, 6, 7, 8, 9, 10 and 11) [8,10,11]. In such cases, the infection spreads between cows, most often by the teat cups, non-sanitized animal equipment, or direct contact between healthy and infected individuals. Immunodepression, deficiency nutrition, infestation of ectoparasites and low biosecurity measures are predisposing factors. The most significant economic factor is associated with secondary infections and losses in milk production. Some lesions can result in skin damage and bleeding, disturb animal welfare, make milking difficult, and predispose the udder to the development of mastitis [12,13]. Cross-infections of BPV are possible in equids, wild and domestic cats, yaks, tapirs, zebras, giraffes, antelopes, buffaloes, sheep, and goats [14,15,16,17,18,19,20,21,22,23]. Until now, transmission to people has not been confirmed. Data from human medicine indicates a high number of human papillomavirus types (over 180) [24], which suggests that the bovine population is still not sufficiently examined and knowledge of BPV pathogenicity and spread seems to be limited.

Little is known about the circulation of BPV in Europe, and no data can yet be found on the prevalence of BPV genotypes that affect the population of Polish cows. This knowledge is valuable as it influences the proper and effective treatment, prophylaxes, and control of the disease. The aim of this study was to genetically evaluate the presence and types of BPV associated with teat papillomatosis in Polish cows. This is also the first report of a new putative type of BPV, type 25, and the so called types BPV42 and BPV43 found in teat papilloma lesions in Europe [4]. BPV42 and BPV43 are named as ‘so called’ types as they have not been classified yet as any particular genus or type and they indicate only a new putative type of isolate name, given by Sauthier et al. in their work from 2021 [4].

## 2. Materials and Methods

### 2.1. Samples and DNA Isolation

The examined samples consisted of skin papillomas collected in the years 2021–2022 from cow teats from the central and western parts of Poland, slaughtered in one of the Polish abattoirs. In total, 177 teat papilloma lesions were sampled from the udder of 109 cows just after slaughter. The lesions were characterized by their morphology, size, and number. The samples for future study were stored directly in a 70% ethanol solution for conservation. DNA was isolated using the Genomic Mini kit (A&A Biotechnology©, Gdańsk, Poland) according to the manufacturer’s instructions and stored at −20 °C for further analysis.

### 2.2. PCR Amplification

PCR amplification was provided based on the major capsid protein (L1) nucleotide sequence of the conserved open reading frame (ORF) region [25]. Two pairs of primers were used: FAP59/FAP64 and MY09/MY11, according to the reaction conditions described earlier (with some modifications) [25,26]. The conditions of the reactions for FAP59/FAP64 were as follows: initial denaturation for 5 min at 95 °C, followed by 35 cycles of 30s denaturation at 95 °C, 30s annealing at 47 °C, 1 min extension at 72 °C, and a final extension at 72 °C for 5 min. For MY09/MY11: initial denaturation at 95 °C for 5 min; the PCR consisted of 35 cycles of 30 s at 95 °C, 30 s at 55 °C and 1 min at 72 °C, followed by extension at 72 °C for 5 min. PCR was performed at a final volume of 25 μL containing: 2.5 μL of 10xDreamTaq™ Green Buffer (Thermo Scientific™, Waltham, MA, USA), 0.1 μL of 100 μM each of the forward and reverse primers (Genomed S.A., Warszawa, Poland), 0.2 μL of 5 U/μL DreamTaq DNA Polymerase (Thermo Scientific™, Waltham, MA, USA), 0.5 μL of 10mM dNTPs Mix (A&A Biotechnology©, Gdańsk, Poland) and 200 ng of the DNA sample. Electrophoresis was performed on 2% agarose gel to see the efficiency of PCR in TBE buffer (A&A Biotechnology©, Gdańsk, Poland) followed by staining with ethidium bromide for 45 min. Products with sizes between 400 and 500 bp were cut from the gel using disposable scalpels and purified with Gel-Out Concentrator kit (A&A Biotechnology©, Gdańsk, Poland) according to the manufacturer’s instructions. 

### 2.3. Sequencing and Phylogenetic Analysis

The fragment obtained from amplification using FAP59/FAP64 primers was chosen for sequencing due to a markedly higher number of obtained products compared to the second pair of primers. The concentration and quality of the purified DNA samples was determined by the DS-11FX Microvolume Spectrophotometer (DeNovix Inc., Wilmington, DE, USA). Isolated DNA fragments were then sent to the external laboratory for both direction sequencing, using the Sanger dideoxy sequencing method (Genomed, Warsaw, Poland and Macrogen Europe, Amsterdam, The Netherlands). Sequences were trimmed and aligned using Chromas software (Technelysium Pty Ltd. ©, South Brisbane, Australia), then compared to sequences already registered in GenBank through the blast server (National Center for Biotechnology Information; http://www.ncbi.nlm.nih.gov/blast/, accessed on 7 March 2023). The isolate with a novel sequence was called a new putative BPV type, since the PCR product was a partial L1 gene nucleotide sequence [27,28]. Some of the obtained sequences were already submitted and registered in GenBank database with the following numbers: ON721392.1 (BPV43), ON721393.1 (BPV8), ON721394.1 (BPV8), ON721402.1 (BPV2), and ON721388.1 (a new putative type). The remaining sequences will be provided during the review process. The phylogenetic tree was constructed using the maximum-likelihood method with 1000 bootstrap replicates through MEGA v.11.0 based on the partial L1 gene nucleotide sequence [29].

## 3. Results

The teat papillomatosis lesions were found in 109 of 800 examined cows (13.6%). In total, 3200 teats were examined, and 177 possessed warts. The lesions were found in 57.4% of cases only on one teat, in 26.7% on two teats, in 10.9% on three teats and in 5.0% in four teats. The observed skin lesions had a different morphology: 83 of the animals had a “rice-grain” shape consisting of a white elongated protuberance, with the diameter oscillating between 0.1 and 1 cm; 6 had cauliflower-like skin papillomas; one filamentous skin papilloma, 1 pedunculated skin papilloma and 18 were mixed in shape (8 “rice-grain” shape and “cauliflower” aspect, 7 “rice-shape” and pedunculated, 2 oval shape and filamentous) (Figure 1 and Figure 2). The average number of changes per teat was 10.8 lesions, and the most infected teat had up to 60 lesions (Figure 3).

Using FAP59/FAP64 primers, 41 positive results were obtained, which constituted 37.6% of the examined animals. However, in the case of the second pair of primers, this number was five. All positive results for the MY09/MY11 primers were also positive for the FAP59/FAP64 primers. A total of 39 positive results were successfully sequenced for the BPV presence. Ten types of BPV were detected, including types 2, 4, 7, 8, 9, 10, 25, the so called isolates BPV42 and BPV43, and one new putative type (isolate PL-25) (Figure 4). Phylogenetic analysis based on the partial major capsid protein L1 gene sequences of BPV isolates from Polish cattle is shown in Figure 5.

## 4. Discussion

Little is known about the prevalence of BPVs in Europe. It is also unclear which of the BPV types are most common in the worldwide population of cattle. Most research identifies types 1 and 2, mainly from skin lesions, and this tendency seems to be accurate for all anatomical regions except for the skin of teats and udder, where type 6 predominates [8,10,11,30,31,32,33,34]. Reports on the prevalence of teat warts are rare. However, the available data show a significant variation in the occurrence of the disease. Low prevalence was reported in Egypt (1.65%), and Bavaria (4.0%), while the disease was relatively common in Bangladesh (15.9%) and Dutch cows, where prevalence decreased from 21.5% to 13.4% after changing of milking system [35,36,37,38]. These results were similar to the present study, where teat wart lesions were found in 13.6% of the examined cows (from which more than one third were positive using PCR amplification). Epidemiological studies provided on cattle in Germany, Italy and Turkey reported ten types of BPV (1, 2, 3, 4, 6, 7, 8, 9, 10, 11) [39,40,41,42,43]. In our study, types 1, 3, 6 (mostly related to teat papillomatosis) and 11 were not detected in examined samples. In Germany, more common infections were correlated with types 8 and 10, while in Italy and Turkey type 1 is the most prevalent (80.0% of examined warts in Italy, and 74.3% in Turkey) [39,40]. In Poland, available data for BPV presence comes only from equine sarcoid examination. BPV confirmed in 40 sarcoids found in 29 horses belonged to type 1, with one exception, where 99% similarity was found between the sequence and BPV type 2 [22]. In our study, infections caused by type 8 and 7 were the most common. Type 8 was reported to be the most common in wart lesions of cows in Germany and Japan [40,44]. Type 7 was first described in skin warts and healthy skin of teats in Japan in 2007, then in cases of teat papillomatosis in Brazil and Germany [8,40,45]. In 2015, its presence in the Italian cattle population was not detected, but viral DNA was found one year later by Savini et al. in the same part of the country as where the earlier work was performed [39,45].

In the literature, PCR amplification of BPV DNA is carried out mostly using degenerate FAP59/FAP64 primers, which were primarily designed to find the L1 nucleotide sequence of the conserved ORF region of human papillomavirus [8,25]. Primers proved to be effective in detecting BPV types 1, 3, 5 and 6 are also MY09/MY11 pair [8,27]. At the beginning of this study, both pairs were used to establish the most efficient and accurate amplification point. In our study, types 10, 25 and so called type 42 and 43 were isolated with the use of MY09/MY11. The use of FAP59/FAP64 primers gave better results. The finding is in agreement with conclusions made by Yıldırım et al. in the paper on BPV detection in teat papillomatosis in Turkey, but other sources show MY09/MY11 as more sensitive [30,46].

As proved before, the type of BPV cannot be guessed just by the location and characteristic features of the lesions. In the literature, “rice-grain” shape lesions were thought to be caused by type 5, but new data show that the same types of lesion can also be seen in infections related to other types, such as BPV 7, 9, 8, 25 and so called type 43 confirmed in our study.

In our study, for the first time, infections caused by type 25 and the so called types BPV42 (potentially classified to the *Xipapillomavirus* genus) and BPV43 (potentially classified to the *Epsilonpapillomavirus* genus) were found in cows in Europe. Outcomes of these types were only reported before in Brazil from cases of teat papillomatosis of milking cows (GenBank: MG252779.1), with low prevalence [4,47]. Macroscopic features of lesions caused by these types have not been described before. In our study, types 25 and 43 characterized only “rice-grain” papillomas, with diameters of 0.2 to 1 cm. Type 42 caused two types of lesions: “rice-grain” with a diameter of 0.3 to 0.5 cm, and cauliflower-like papilloma (2.5 cm length). Additionally, 40% similarity between the detected type BPV42 was found with two human papillomaviruses isolated from healthy skin (GenBank number: KC752024.1) and cases associated with epidermodysplasia verruciformis in children (GenBank number: AY044281.1), respectively, coming from China and France [48]. In our study, a new putative BPV type was found (isolate PL-25), showing 75% similarity with already registered in GenBank database isolates BPVUFSBR-45 (GenBank number: MT674614.1) and BPVUFSBR-34 (GenBank number: MT674603.1), isolated in Brazil by Figueirêdo et al. [47]. These isolates were phylogenetically closer to type 25, but PL-25 shows more correlation with the so called type BPV43 and may potentially belong to the *Epsilonpapillomavirus* genera (Figure 5). Macroscopically, the lesions caused by PL-25 isolate were “rice-grain” in shape, and of a pedunculated form, with the infection spread over three teats. These findings show that the situation is dynamically changing, and the spread of new types can be seen in European herds through the years [4]. Even in different regions of the world, the tendencies may be comparable, as described between Poland and Japan [44].

Knowledge of most common BPV types allows the management of the disease in the examined population of cows. The most severe and resistant to treatment papillomatosis is caused by *Deltapapillomaviruses* (BPV-1, 2, 13, 14). The severity depends on the presence of three oncogenic proteins in the viral genome—E5, E6 and E7, and two of them are usually sufficient to successfully develop malignant tumor lesions [5]. An exception is type 8, which possesses all the oncogenic proteins but is related to mild and self-limited lesions [5]. BPV types detected in our study belong to four genera: *Delta*-, *Xi*-, *Dyoxi*- and *Epsilonpapillomavirus*. Only type 2 from *Deltapapillomavirus* genera could be isolated from teat papillomatosis cases (Figure 3). What is interesting is that teats with the highest number of lesions were correlated with type 8 infections (Figure 3). This may suggest that, although the course of the disease caused by this type may be benign and the lesions self-limited, the virus has the potential to spread more intensively and cause more numerous lesions than other types, which makes treatment difficult, especially for lesions found on teats.

To date, no agreement has been reached on one efficient and sensitive method for BPV genotyping [40]. Many reports from recent years show that co-infections of at least two BPVs in one lesion are possible. In Italy, Grindatto et al. found this correlation in 2 of 70 sampled warts, with coinfection of BPV-1 and BPV-3, and BPV-1 and BPV-9 [39]. In our study, two of the samples with strong products in the electrophoresis were not useful for sequencing, and we suppose the reason for this was multi-type infections. We were unable to confirm the findings using the methodology of this study. Further studies of gained isolates using molecular cloning should be performed to confirm the possibility of BPV mixed infections. The PCR product should be cloned into plasmid, then multiple clones should be sequenced [39].

Despite the characteristic morphology of the lesions, we were not able to amplify viral DNA in the rest of the examined teats samples. The problem might have been generated because the lesions were too old and the viral DNA could not be longer found there, or because of the viral DNA integration with the host genome, which is the natural mechanism for BPV infections [5,49].

## 5. Conclusions

The research concerning BPV infections focuses mostly on skin lesions of the whole body, except for the skin of teats and udder. In practice, lesions found on teats seem to be more important as they directly influence losses in milk production, and treatment possibilities are usually limited because of the specific anatomic region. Our study has shown that teat papillomatosis in Poland constitutes a relevant problem in dairy cattle production, with one of the highest prevalences reported in the literature.

Our study has shown a wide variety of BPV types that cause teat and udder papillomatosis in Polish cows. Moreover, we detected a new putative BPV type and BPV types not reported before in the European cattle population (type 25, and the so called types 42 and 43). Among identified viruses, there are two BPV genotypes (type 7 and 8) occurring significantly more frequently in Polish cattle. We can also conclude that morphology of the lesions observed is not correlated with the specific BPV type.

The BPV shows a unique ability for the *Papillomaviridae* family to cross inter-species boundaries, and better study of its characterization in addition to the clinical and epidemiological aspects should be conducted [14,15,16,17,18,19,20,21,22,23,50]. Viral DNA is found in healthy (latent form of infection) and sick animals in the lymph nodes, blood, sperm, saliva, urine, placenta, fetuses, and milk after pasteurization [5,6,7,8,9,51]. Therefore, further research into its specifications and the potential for transmission to humans is crucial to protect public and animal health.

## Figures and Tables

**Figure 1 pathogens-12-01278-f001:**
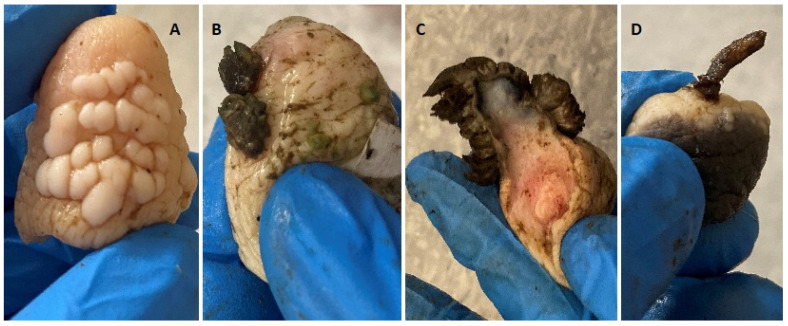
Skin papillomas’ macroscopic view, found in examined population of Polish cows. (**A**)—“rice-grain” shape consisting of a white elongated protuberance, (**B**)—Cauliflower-like skin papilloma, (**C**)—Filamentous skin papilloma, (**D**)—Pedunculated skin papilloma.

**Figure 2 pathogens-12-01278-f002:**
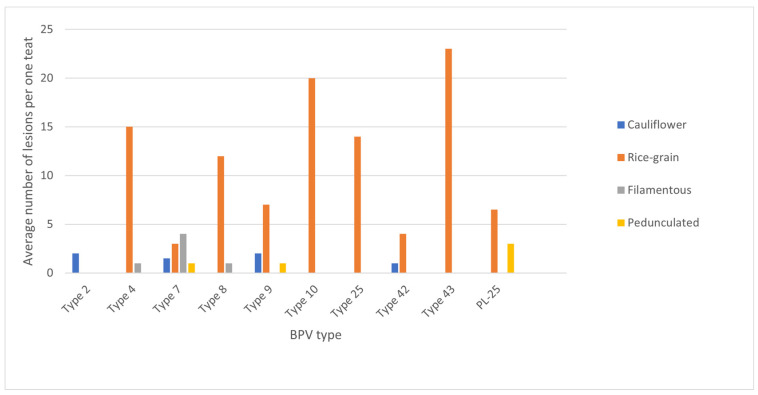
Average number of lesions per teat depending on the macroscopic feature and BPV type isolated from the lesions.

**Figure 3 pathogens-12-01278-f003:**
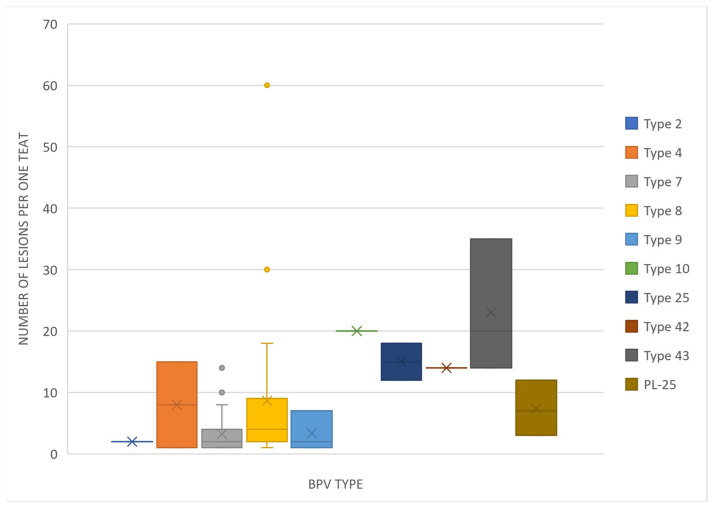
Box-and-whisker plot showing the number of lesions found on one teat depending on the BPV type isolated.

**Figure 4 pathogens-12-01278-f004:**
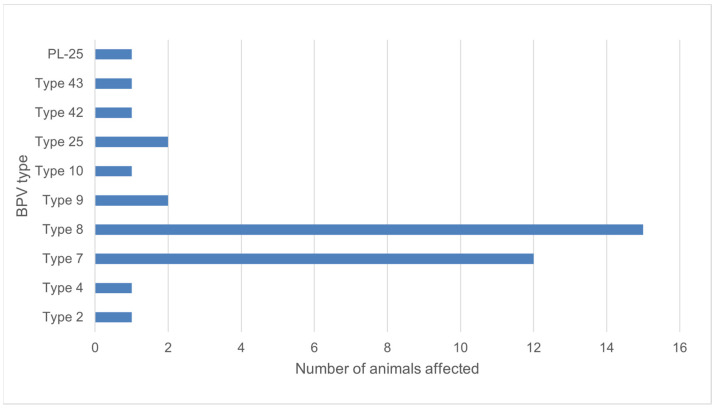
Number of animals affected by isolated BPV types.

**Figure 5 pathogens-12-01278-f005:**
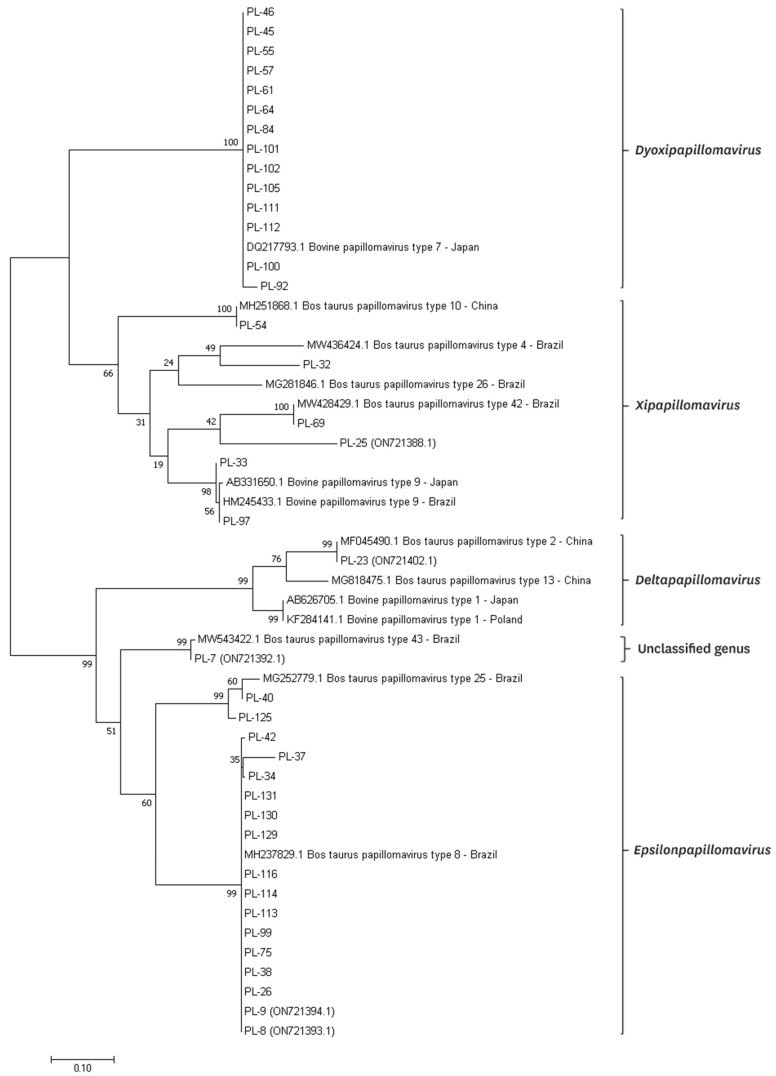
Phylogenetic analysis based on the partial major capsid protein L1 gene sequences of BPV isolates from Polish cattle. Phylogenetic tree was generated by the maximum-likelihood method with 1000 bootstrap replicates through the MEGA v11.0 software.

## Data Availability

No new data were created or analyzed in this study. Data sharing is not applicable to this article.

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
