# Peer review of "Genetic Evaluation of Bovine Papillomavirus Types Associated with Teat Papillomatosis in Polish Dairy Cattle with the Report of a New Putative Type"

_pathogens, 2023, doi:10.3390/pathogens12111278_

Round 1

Reviewer 1 Report

Comments and Suggestions for Authors

I found the article well written, interesting, well structured and easy to read.

Reviewer 2 Report

Comments and Suggestions for Authors

The authors have statistically reported the prevalence of BPV by morphological recording and genetic evaluation. This is a concise manuscript which delivers significant epidemiological data about the confirmed as well as putative types of BPV prevalent in Poland, which have caused economic losses yet never reported before in Poland. Overall, this manuscript is prepared well. Below I listed some minor points for the authors’ consideration.

1 In page 1, line 29, “specie-specific” should be replaced with “species-specific”.

2 In Figure 3, you should change “New putative type” to “PL-25” in order to make it consistent with other figures and interpretations in manuscript.

3 In page 8, lines 201-202, PL-25 caused filamentous form of lesions, while in Figure 3, such type of BPV caused pedunculated lesions.

4 There was no “Table 1” in this manuscript, so in page 8, line 216, “Table 1” might need to be replaced with “Figure 2”.

Reviewer 3 Report

Comments and Suggestions for Authors

The authors identified a novel BPV type and documented epidemiology of BVP among Polish dairy cattle. This is a well-written paper containing interesting results which merit publication. For the benefit of the readers, however, a number points require further justification. These are given below.

1. Line 24. Delete BPV.

2. Line 28 and others. Family name and genus name should be in Italic as in line 209.

3. Line 51 and many others. BPV not bovine papilomavirus.

4. Line 57. The authors should explain what "so-called" means in the introduction section.

5. Line 69. The major capsid protein (L1) nucleotide sequence.

6. Line 73. Follows not following.

7. Line 89. Markedly not significantly.

8. Figs. 2 and 4. The results should be shown on box-and-whisker plots.

9. Fig. 5. It's better to include genus name in the figure.

10. Lines 152-153. Present not presented.

11. Line 171 . L1 nucleotide sequence.

13. Lines 185 and 186. Genus not genera.

14. Lines 204-206. This sentence needs citation.

15. Lines 224-226. The PCR product should be cloned into plasmid, then multiple clones should be sequenced.

Comments on the Quality of English Language

There are a number of errors as shown above.

Round 2

Reviewer 3 Report

Comments and Suggestions for Authors

The manuscript has been improved. I have no serious comment.